# Assessing the Predictive Accuracy of the Eaton–Littler Classification in Thumb Carpometacarpal Osteoarthritis: A Comparative Analysis with the Outerbridge Classification in a Cohort of 51 Cases

**DOI:** 10.3390/diagnostics14161703

**Published:** 2024-08-06

**Authors:** Enrico Carità, Alberto Donadelli, Mara Laterza, Giacomo Rossettini, Jorge Hugo Villafañe, Pier Giuseppe Perazzini

**Affiliations:** 1Clinica San Francesco, Via Monte Ortigara 21, 37127 Verona, Italy; drcaritaenrico@gmail.com (E.C.); albertodonadelli@gmail.com (A.D.); mara.laterza85@gmail.com (M.L.); eppeperazzini@gmail.com (P.G.P.); 2Department of Physiotherapy, Faculty of Sport Sciences, Universidad Europea de Madrid, 28670 Villaviciosa de Odón, Spain; mail@villafane.it

**Keywords:** thumb, osteoarthritis, arthroscopy

## Abstract

(1) Background: The objective of this study is to evaluate the predictive value of the Eaton–Littler radiologic classification for thumb carpometacarpal osteoarthritis (CMC OA) relating to intra-articular cartilage damage assessed by the Outerbridge arthroscopic classification. (2) Methods: A total of 51 thumb CMC OA arthroscopies were performed on patients classified as Eaton stages 1, 2, or 3. Post-arthroscopic evaluations of cartilage damage were categorized using the Outerbridge classification. Comparative analyses were conducted between the radiological Eaton stages and the arthroscopic Outerbridge stages. (3) Results: Arthroscopic examination revealed Outerbridge stage 3 and 4 cartilage damage in 26 cases classified as Eaton stage 2 and in 18 cases classified as Eaton stage 3. The detection of severe cartilage damage in patients classified as Eaton stage 2 was unexpected. (4) Conclusions: Arthroscopy demonstrated that many patients with mild radiological degenerative signs exhibited significant cartilage destruction. Although the Eaton classification is widely used for staging thumb CMC OA, it may not accurately reflect the severity of intra-articular damage. The Eaton classification does not reliably predict intra-articular damage in Eaton stage 2 cases.

## 1. Introduction

Thumb carpometacarpal osteoarthritis (CMC OA) is a degenerative alteration of the CMC joint, which is characterized by abrasion, progressive deterioration of the joint surfaces, ligamentous laxity, and osteophyte formation at the site of damage [1]. The condition affects approximately 15% of adults older than 30 years and up to 66% of women older than 55 years; it results in a substantial impairment of the upper extremity’s function [2]. Despite the high prevalence, the variability in pain caused by thumb CMC OA can be significant, and radiographic severity does not reliably predict disability [3]. This discrepancy is likely linked to the level of inflammation in the joint and its synovial tissue, explaining why some patients with severe OA are asymptomatic while others with mild radiological signs experience severe pain and loss of strength [4]. Pain and loss of pinch strength are the main symptoms limiting a patient’s activities and reducing their quality of daily life [5,6].

Conservative treatment is commonly indicated in the early stages of OA but often yields unpredictable results. Recent systematic reviews have highlighted the inconclusive evidence regarding the effectiveness of conservative interventions such as orthoses, exercise, manual therapy, and other modalities [7]. It has been shown that while conservative approaches are the initial choice, their efficacy in alleviating pain and improving function remains uncertain [8,9].

For patients who do not respond adequately to conservative treatment modalities, there is a myriad of surgical techniques available. The surgical options include trapeziectomy alone or with a concomitant soft tissue procedure (ligament reconstruction and/or tendon interposition or suspension), arthrodesis, and different types of prosthetic implants [10].

Despite the high prevalence and significant impact of thumb CMC OA, the pathophysiological mechanisms driving the variability in clinical presentation remain poorly understood. For patients who do not respond adequately to conservative treatment modalities, there is a myriad of surgical techniques available. The surgical options include trapeziectomy alone or with a concomitant soft tissue procedure (ligament reconstruction and/or tendon interposition or suspension), arthrodesis, and different types of prosthetic implants. The interplay between mechanical stress, cartilage degeneration, and synovial inflammation is complex, contributing to the challenge of predicting patient outcomes based on radiographic findings alone. The Eaton–Littler classification is widely used for radiological staging of thumb CMC OA, categorizing the disease into four stages based on the extent of joint space narrowing, subluxation, and osteophyte formation (Table 1 and Figure 1) [11]. However, this classification system does not account for the extent of intra-articular cartilage damage, which can vary significantly among patients with similar radiographic stages. To address this gap, the Outerbridge classification system provides a framework for assessing intra-articular cartilage damage through arthroscopic evaluation (Table 2) [12]. This system categorizes cartilage lesions from normal (Stage 0) to severe erosion exposing subchondral bone (Stage 4). By comparing the Eaton–Littler radiologic stages with the Outerbridge arthroscopic findings, this study aims to evaluate the correlation between these two assessment methods in patients with symptomatic thumb CMC OA. 

This study hypothesizes that the extent of articular damage in thumb CMC OA does not correspond well with radiological staging, which is commonly used for assessment. We aimed to evaluate the correlation between radiological staging, using the Eaton–Littler classification, and intra-articular cartilage damage, assessed via the Outerbridge classification, through arthroscopic evaluation in patients with symptomatic thumb CMC OA. This comparative analysis seeks to determine whether radiological assessments accurately reflect the severity of intra-articular damage and to identify any discrepancies that may affect clinical management and outcomes.

By assessing the predictive value of the Eaton–Littler radiologic classification for thumb CMC OA relating to intra-articular cartilage damage assessed by the Outerbridge arthroscopic classification, this study aims to provide a more accurate diagnostic tool for clinicians. Previous research has indicated a significant variation in cartilage damage among patients with similar radiographic stages, suggesting the need for a comprehensive classification system that includes both radiographic and clinical parameters.

## 2. Materials and Methods

### 2.1. Study Design

This retrospective study aims to assess the predictive accuracy of the Eaton–Littler radiologic classification [12] for thumb CMC OA by comparing it with the Outerbridge arthroscopic classification [13,14]. The goal is to determine the extent of correlation between radiologic staging and intra-articular cartilage damage assessed through arthroscopy.

### 2.2. Patient Selection

From January 2018 to January 2023, a total of 51 patients underwent arthroscopy for symptomatic thumb CMC OA before fat micrograft articular injection. Patients were included in the study based on specific criteria to ensure the relevance and accuracy of the findings. The inclusion criteria required patients to be diagnosed with chronic stage I, II, or III basal joint arthritis according to the Eaton–Littler classification and to have experienced symptoms for at least six months. Both male and female patients were considered eligible. 

Patients were excluded if they had a history of CMC trauma, rheumatic diseases, CMC instability or subluxation, hyper-laxity, recent injections of hyaluronic acid, PRP, or corticosteroids, or recent treatment with shock waves. In cases where both hands were affected, the more symptomatic hand was included in the study to maintain consistency in the data.

### 2.3. Preoperative Assessment

The arthroscopic procedures were performed under local anesthesia and sedation to ensure patient comfort and minimize discomfort. The patients underwent arthroscopy for both diagnostic purposes and to perform concomitant treatments. Patients were positioned supine on an operating table, with the arm stabilized on a standard arm table and a tourniquet applied to control blood flow. A single Chinese finger trap was used to apply longitudinal traction to the thumb while an assistant held the forearm in pronation.

### 2.4. Operative Procedure

The procedure involved several key steps to ensure accurate and thorough assessment. The surgery was conducted under local anesthesia and sedation. The arthroscopic part of the surgery lasted about 5 min. Patients tolerated the tourniquet well with the aid of sedation: 

Portal Identification: The thumb CMC joint was located by palpation, and a 22-gauge needle was inserted ulnar to the extensor pollicis brevis tendon to identify the 1-U portal. A No. 11 blade was used to incise the skin, and a blunt hemostat facilitated reaching the joint.

Arthroscope Insertion: A 1.9-mm arthroscope with a 30° inclination was inserted into the basal joint to provide a complete visualization of the joint surfaces, capsule, and ligaments.

Second Portal: Another 22-gauge needle was inserted into the basal joint from the skin overlying the 1-R portal location, radial to the abductor pollicis longus tendon.

Joint Assessment: The metacarpal and trapezial surfaces were assessed using a small probe, ensuring a thorough examination of the joint’s condition.

Closure: The procedure concluded with skin sutures using No. 5-0 monofilament, followed by the application of a sterile bandage.

### 2.5. Data Collection

All arthroscopic procedures were recorded to ensure accurate and detailed evaluation. Two independent observers classified the cartilage damage using the Outerbridge classification system. The final decision in case of disagreement was made by the senior observer with more experience. Both observers staged the X-rays, and the staging was not blinded. In cases where different stages of cartilage degeneration were observed within the same joint, the worst grade of damage with a surface area greater than 2 mm was used for classification. This approach ensured that the most severe condition was recorded for each joint, providing a reliable basis for comparison with radiologic findings.

### 2.6. Statistical Analysis

Descriptive statistics were used to analyze the data, providing a clear overview of the findings. The correlation between the Eaton–Littler radiologic stages and the Outerbridge arthroscopic stages was assessed using statistical tests to ensure the robustness of the conclusions:

Fisher’s Exact Test: This test was used to determine the significance of the correlation between radiologic and arthroscopic findings.

Spearman’s Rank Correlation Coefficient: This coefficient was employed to evaluate the strength and direction of the association between radiologic stages and the severity of cartilage damage. The raw data supporting the conclusions of this article will be made available by the authors on request.

### 2.7. Ethical Considerations

The study was conducted in strict accordance with ethical guidelines to ensure the protection and respect of patient rights. All patients provided informed consent for the use of their data in this study, acknowledging their understanding and agreement to participate. Additionally, the study received approval from the Institutional Review Board of Garofalo Health Care GHC (protocol code 0014GHCIRB), approved on 30 May 2024, ensuring that all procedures were ethically sound and in line with the Declaration of Helsinki.

## 3. Results

A total of 51 patients were included in this study: 12 men and 39 women. The mean age was 55.5 years (72 max and 27 min). All patients had radiographs consisting of posteroanterior, lateral, and Robert views of the thumb.

Based on X-rays, two patients had stage I thumb OA, 31 patients had stage II, and 18 patients had stage III (Table 3). 

Based on the arthroscopic check, two patients had stage 1 Outerbridge, five patients had stage 2, 21 patients had stage 3, and 23 patients had stage 4 (Table 4).

According to the Eaton and Outerbridge classifications, the two patients that belong to the Eaton stage I group had stage 1 Outerbridge. Of the 31 patients that belong to the Eaton stage II group, 0 patients had stage 1 Outerbridge, five patients had stage 2, 14 patients had stage 3, and 12 patients had stage 4. Of the 18 patients that belong to the Eaton stage III group, seven patients had stage 3 Outerbridge, and 11 patients had stage 4.

No complications were detected after surgery.

Statistical analysis showed a *p*-value of 0.002 at the Fisher test, demonstrating that in advanced radiological classification stages, we can find the worst cartilage damage. The Spearman test had a *p*-value of 0.0075, showing that there is a mild positive correlation between radiological and cartilage damage (Table 5).

## 4. Discussion

The Eaton and Littler classification for thumb basilar joint arthritis is based solely on radiographic parameters, regardless of qualitative clinical findings. This classification describes the progression of thumb CMC OA in only four stages according to narrowing and osteophyte dimensions, simplifying a complex degenerative process in the second and third stages and disregarding relevant radiological factors that correlate to the state and pitfalls of the joint like stability or bone morphology. Despite its excessive simplicity, a poor intra and inter-observer reliability of this staging system has been published [15], which can explain why the radiographic severity of thumb basilar joint arthritis has not been found to correlate with the severity of symptoms in all cases [16]. On the contrary, because of the more objective radiographic nature of the classification, the Eaton-Litter classification has been considered useful in clinical practice and has gained widespread acceptance [17]. Multiple studies demonstrate large variations in the utility of this classification; furthermore, an absence of agreement exists for correlating recommended treatment to CMC radiographic severity [18,19,20]. 

In common practice, treatment recommendations are often based on radiographic findings in conjunction with symptoms and clinical examination [21]. However, this matching becomes challenging in patients who report severe symptoms of thumb basilar joint arthritis despite having limited radiographic evidence of such. In this regard, current literature suggests that the Eaton–Littler stage does not correlate well with clinical findings, and treatment depends on patient response to nonoperative measures. A review indicated that patients with stage I osteoarthritis are likely to benefit from conservative management, and the choice of treatment methods for those with stage II, III, and IV osteoarthritis depends on the severity of patient symptoms and their functional demands [22].

To our knowledge, in the literature, there is no study about the correlation of Outerbridge classification and symptom severity in thumb CMC OA.

In this study, we describe how radiographs cannot predict the real severity of actual cartilage degeneration at the thumb CMC joint. In our findings, patients with Eaton–Littler stage II have a wide variability of Outerbridge classification, and 19 of 29 patients (65%) had severe chondral alterations classified by Outerbridge stages 3 and 4, therefore an advanced stage (Figure 2).

Villafañe et al. [23,24] found that radial nerve mobilization can reduce pain sensitivity and improve motor performance in patients with thumb CMC OA, emphasizing the importance of considering factors beyond simple radiographic evaluation. Additionally, recent literature suggests that the gut microbiota may play a crucial role in mediating osteoarthritis through systemic inflammation, offering new avenues for research in managing OA pain [25].

In our study, the presence of many cases of severe cartilage damage in Eaton stage 2, which is commonly considered an early stage, suggests two considerations: the first confirms that radiological classification underestimates the effective degenerative stage of the CMC joint; the second can explain why patients with early Eaton–Littler stage but poor effective cartilage conditions sometimes experience persistent pain after conservative treatment [26]. There might be a diagnostic gap in Eaton stage 2 that does not allow for the proper classification and treatment of patients in this stage. More sensitive methods than radiographs are needed to better define the presence and severity of thumb CMC arthritis. CT and CT CONE BEAN scans could be more effective in joint degeneration staging showing geods, isolated narrowing, and loose bodies and improving the accuracy of radiological staging. 

### 4.1. Study Limitations

Several limitations must be acknowledged in this study. It is a small retrospective series of patients already selected for surgical treatment, and the same surgeon performed both the radiological and arthroscopic staging. Clinically, cartilage degeneration is not the sole factor leading to symptomatic osteoarthritis, and the correlation between the extent of cartilage damage and symptoms remains unclear. Additionally, factors such as bone edema and arthrosynovitis, which can contribute to CMC joint pain, were not accounted for and are not detectable by X-ray or arthroscopy. Moreover, no statistical correlation was identified between the Eaton–Littler and Outerbridge classifications. 

### 4.2. Clinical Implications

Developing a more comprehensive classification system that integrates both radiographic and clinical parameters is crucial for the effective management of thumb carpometacarpal (CMC) osteoarthritis (OA). The findings from our study highlight several key clinical implications that could significantly impact the diagnosis, treatment, and management of this condition.

The limitations of the Eaton–Littler classification in accurately reflecting the severity of cartilage damage underscore the need for advanced imaging techniques. Standard radiographs often fail to capture the true extent of joint degeneration, particularly in the early stages. Advanced imaging modalities such as CT (Computed Tomography) and Cone Beam CT scans can provide a more detailed visualization of the joint structure, revealing critical aspects like geodes, isolated joint space narrowing, and loose bodies. These techniques allow for a more precise assessment of the joint, enabling clinicians to make more informed decisions regarding the appropriate intervention strategies.

A significant gap exists between radiographic findings and clinical symptoms in patients with thumb CMC OA. For instance, some patients with mild radiographic changes may experience severe symptoms, while others with advanced radiographic stages may remain relatively asymptomatic. This discrepancy suggests that relying solely on radiographic staging is insufficient. A comprehensive classification system should incorporate clinical parameters such as pain severity, functional impairment, and physical examination findings [27]. This integrated approach can provide a more accurate depiction of the disease state, leading to better-targeted treatments.

The variability in cartilage damage among patients classified under the same Eaton–Littler stage indicates that treatment plans need to be personalized. For example, patients with Eaton stage II but severe cartilage damage (Outerbridge stage 3 or 4) may require more aggressive interventions than those with similar radiographic findings but less severe cartilage damage. Personalized treatment plans should be developed based on a combination of radiographic data, clinical symptoms, and patient-specific factors such as activity level and overall health status. This tailored approach can enhance the effectiveness of both conservative and surgical treatments.

Conservative management remains the first line of treatment for early-stage thumb CMC OA. However, the unpredictability of its outcomes calls for enhanced strategies that go beyond traditional methods. Incorporating advanced diagnostic tools to better understand the extent of cartilage damage can help tailor conservative treatments more effectively. Techniques such as radial nerve mobilization, as highlighted by Villafañe et al., show promise in reducing pain and improving motor performance. Additionally, exploring the role of systemic factors like gut microbiota in OA can lead to innovative conservative management approaches that address the underlying inflammatory processes.

Early identification of patients who are likely to progress to more severe stages of thumb CMC OA is essential for preventing significant joint damage and preserving function. The current study indicates that severe cartilage damage can occur even in early radiographic stages. Therefore, clinicians should consider incorporating advanced imaging and comprehensive clinical evaluations early in the diagnostic process. Early intervention strategies, including targeted physical therapy, lifestyle modifications, and potentially pharmacological treatments, can slow the progression of the disease and improve long-term outcomes.

To implement these advanced diagnostic and treatment approaches effectively, educational and training programs for healthcare providers are necessary. Training programs should focus on the interpretation of advanced imaging techniques, the integration of clinical findings with radiographic data, and the development of personalized treatment plans. By enhancing the skill set of clinicians, these programs can ensure that patients receive the most accurate diagnoses and effective treatments.

The insights gained from this study pave the way for future research aimed at improving the classification and management of thumb CMC OA. Further studies should validate the use of advanced imaging techniques and explore new diagnostic criteria that combine radiographic and clinical parameters. Research should also investigate the efficacy of various conservative and surgical interventions based on more accurate staging systems. Additionally, exploring the biological mechanisms underlying the disease, such as the role of gut microbiota, can lead to novel therapeutic targets.

## 5. Conclusions

This study reveals that the Eaton–Littler classification often underestimates the severity of cartilage damage in thumb CMC OA, especially in Eaton stage 2, where severe damage (Outerbridge stages 3 and 4) frequently goes undetected radiographically. Incorporating advanced imaging techniques like CT or Cone Beam CT scans is advisable for a more accurate cartilage assessment and staging. The significant variability in damage among patients with similar radiographic stages suggests the need for a comprehensive classification system that includes both radiographic and clinical parameters. Future research should validate these findings and explore additional diagnostic criteria to improve the management of thumb CMC OA.

While the Eaton–Littler classification has been a mainstay in the radiographic assessment of thumb CMC OA, its limitations are evident. A more nuanced approach that incorporates advanced imaging and considers clinical symptoms alongside radiographic findings is necessary for a holistic understanding and effective management of the disease. The development of such an approach will likely lead to better patient outcomes and more targeted therapeutic interventions. Further studies should focus on refining classification systems and exploring new diagnostic modalities that can bridge the gap between radiographic findings and clinical reality, ultimately enhancing the care provided to patients with thumb CMC OA.

## Figures and Tables

**Figure 1 diagnostics-14-01703-f001:**
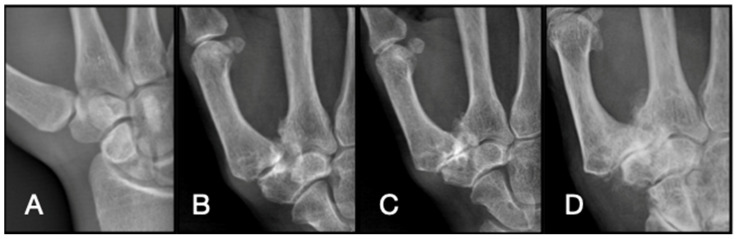
Eaton–Littler Classification. Stage I (**A**), II (**B**), III (**C**), and IV (**D**) carpometacarpal arthrosis of the Eaton–Littler classification system.

**Figure 2 diagnostics-14-01703-f002:**
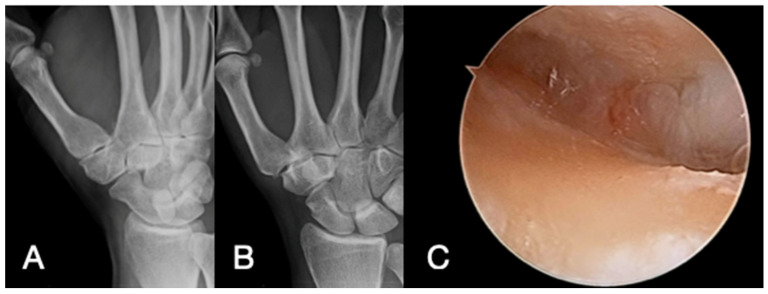
Clinical case of a 47-year-old woman with symptomatic right thumb osteoarthritis. X-rays (**A**,**B**) show Eaton–Littler stage II, while arthroscopic image (**C**) shows a chondral lesion as Outerbridge stage 4.

**Table 1 diagnostics-14-01703-t001:** Eaton–Littler Classification.

Stage	Description
I	Slight carpometacarpal joint space widening, normal articular contours; less than 1/3 subluxation in any projection
II	Slight carpometacarpal joint space narrowing, sclerosis, and cystic changes with osteophytes or loose bodies <2 mm, usually adjacent to the volar or dorsal facets of the trapezium; at least 1/3 subluxation of the joint
III	Advanced carpometacarpal joint space narrowing, sclerosis, and cystic changes with osteophytes or loose bodies >2 mm, dorsally or volarly, usually in both locations; greater than 1/3 subluxation
IV	Advanced degenerative changes; major subluxation and very narrow joint space, with cystic and sclerotic subchondral bone changes with scaphotrapezial arthritis

**Table 2 diagnostics-14-01703-t002:** Outerbridge Classification.

Stage	Description
0	Normal cartilage
1	Chondral lesions are characterized by softening and swelling, which often require tactile feedback with a probe or other instrument to assess
2	Partial-thickness defect of cartilage with fissures that do not exceed 0.5 inches in diameter or reach subchondral bone
3	Fissuring of the cartilage with a diameter >0.5 inches with an area reaching the subchondral bone
4	Erosion of the articular cartilage that exposes subchondral bone

**Table 3 diagnostics-14-01703-t003:** Radiologic Stage distribution according to Eaton-Litter Classification.

Stage	Frequency	Percentage
I	2	3.9%
II	31	60.8%
III	18	35.3%
Total	51	100%

**Table 4 diagnostics-14-01703-t004:** Arthroscopic Stage distribution according to Outerbridge Classification.

Stage	Frequency	Percentage
1	2	3.92%
2	5	9.80%
3	21	41.18%
4	23	45.10%
Total	51	100%

**Table 5 diagnostics-14-01703-t005:** Arthroscopic and Radiological Stage distribution.

Radiological Stage	ARS * 1	ARS 2	ARS 3	ARS 4	Total
I	2 100%	0	0	0	2
II	0	5 16.13%	14 45.16%	12 38.71%	31
III	0	0	7 38.89%	11 61.11%	18
Total	2 3.92%	5 9.80%	21 41.18%	23 45.10%	51 100%

* Arthroscopic Stage.

## Data Availability

The raw data supporting the conclusions of this article will be made available by the authors on request.

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
