# Peer review of "Assessing the Predictive Accuracy of the Eaton–Littler Classification in Thumb Carpometacarpal Osteoarthritis: A Comparative Analysis with the Outerbridge Classification in a Cohort of 51 Cases"

_diagnostics, 2024, doi:10.3390/diagnostics14161703_

Round 1

Reviewer 1 Report

Comments and Suggestions for Authors

Thank you for letting me review this article.

The topic is very interesting, the study is well conducted, and the article is well writen.

A few minor modifications could imorive it making it more clear, especify clarifying some methodological doubts:

Line 34-36: this sentence is useless in the introduction of this study.

Line 48-49 this sentence is repeated exactly 5 lines above (42-43). Delete one of the sentences to avoid repetition… same problem for the following sentence.

Line 53-58 you discuss conservative management of CMC OA that is not directly relevant to the study… but you do not mention surgical treatment. Either you delete this paragraph, or you should at list briefly discuss surgical possibility.

Line 111-116 specify why does patients underwent arthroscopy: was it diagnostic, or did they had a concomitant treatment?

Line 123 was it local anesthesia or loco-regional (plexus)… if it was local how long did the procedure last, was the tourniquet an issue?

Line 143-149:

- if the two independent observers did not agree, who made the final decision?

- who staged the x-rays? 

- where the two different staging blinded?

(Line 244 you say 1 surgeon performed both stagings… decide which version is the correct one)

Line 175 you are missing a “,” between “stage 2 “and “21 patients”

Line 182-185: edit English

Line 217-221 it would be interesting -if it exists- to cite a study that correlates Outerbridge classification with symptoms and discuss it. If it does not exist you should also point that out.

Line 225 how these statements is related to Outerbridge classification? I would think about deleting this paragraph. Moreover same concepts are discussed line 281-288 where they fit better.

Line 254 write directly CMC OA, you do not need to write it in extended version.

Comments on the Quality of English Language

see comments above (just a small paragraph does not sound good)

Author Response

REVIEWER 1

Thank you for letting me review this article.

The topic is very interesting, the study is well conducted, and the article is well writen.

A few minor modifications could imorive it making it more clear, especify clarifying some methodological doubts:

Line 34-36: this sentence is useless in the introduction of this study.

Response: We agree. We removed it.

Line 48-49 this sentence is repeated exactly 5 lines above (42-43). Delete one of the sentences to avoid repetition… same problem for the following sentence.

Response: We removed it.

Line 53-58 you discuss conservative management of CMC OA that is not directly relevant to the study… but you do not mention surgical treatment. Either you delete this paragraph, or you should at list briefly discuss surgical possibility.

Response:  We added a small paragraph about surgical options.

Line 111-116 specify why does patients underwent arthroscopy: was it diagnostic, or did they had a concomitant treatment?

Response: We did it.

Line 123 was it local anesthesia or loco-regional (plexus)… if it was local how long did the procedure last, was the tourniquet an issue?

Response: We did surgery under local anesthesia and sedation. The arthroscopic part of surgery lasted about 5 minutes. Patients tolerated tourniquet aided by sedation.

Line 143-149:

- if the two independent observers did not agree, who made the final decision?

Response: The senior one, who has major experience.

- who staged the x-rays?

Response: The two observes.

- where the two different staging blinded?

Response: No.

(Line 244 you say 1 surgeon performed both stagings… decide which version is the correct one)

Response: It was a mistake, we removed it.

Line 175 you are missing a “,” between “stage 2 “and “21 patients”

Response: We added “,”

Line 182-185: edit English

Response: We corrected it.

Line 217-221 it would be interesting -if it exists- to cite a study that correlates Outerbridge classification with symptoms and discuss it. If it does not exist you should also point that out.

Response: We corrected it.

Line 225 how these statements is related to Outerbridge classification? I would think about deleting this paragraph. Moreover same concepts are discussed line 281-288 where they fit better.

Response: We corrected it.

Line 254 write directly CMC OA, you do not need to write it in extended version.

Response: We corrected it.

Reviewer 2 Report

Comments and Suggestions for Authors

Dear Authors,

The subject of the manuscript is very interesting. I believe that the authors presented the subject correctly and thoroughly the introduction is adequate, as ate the others parts of the manuscript.

However, I believe that there are some limitations of the study related to the relatively small number of patients  included in the study and on which the

evaluations were carried out. This fact should perhaps be specified.

Best regards,   

Author Response

REVIEWER 2

Dear Authors,

The subject of the manuscript is very interesting. I believe that the authors presented the subject correctly and thoroughly the introduction is adequate, as ate the others parts of the manuscript.

However, I believe that there are some limitations of the study related to the relatively small number of patients  included in the study and on which the

evaluations were carried out. This fact should perhaps be specified.

Best regards,

Response: Thank you for your comment. We added this observation.